# Biocompatibility of Root Canal Sealers: A Systematic Review of In Vitro and In Vivo Studies

**DOI:** 10.3390/ma12244113

**Published:** 2019-12-09

**Authors:** Diogo Afonso Fonseca, Anabela Baptista Paula, Carlos Miguel Marto, Ana Coelho, Siri Paulo, José Pedro Martinho, Eunice Carrilho, Manuel Marques Ferreira

**Affiliations:** 1Institute of Endodontics, Coimbra Institute for Clinical and Biomedical Research (iCBR), CIBB Center for Innovative Biomedicine and Biotechnology, Faculty of Medicine, University of Coimbra, 3000-075 Coimbra, Portugal; 2Institute of Integrated Clinical Practice, Coimbra Institute for Clinical and Biomedical Research (iCBR), CIBB Center for Innovative Biomedicine and Biotechnology, CIMAGO—Center of Investigation on Environment, Genetics and Oncobiology, CNC.IBILI, Faculty of Medicine, University of Coimbra, 3000-075 Coimbra, Portugal; anabelabppaula@sapo.pt (A.B.P.); mig-marto@hotmail.com (C.M.M.); anasofiacoelho@gmail.com (A.C.); eunicecarrilho@gmail.com (E.C.); 3Institute of Experimental Pathology, Faculty of Medicine, University of Coimbra, 3000-075 Coimbra, Portugal; 4Institute of Endodontics, Coimbra Institute for Clinical and Biomedical Research (iCBR), CIBB Center for Innovative Biomedicine and Biotechnology, CIMAGO – Center of Investigation on Environment, Genetics and Oncobiology, CNC.IBILI, Faculty of Medicine, University of Coimbra, 3000-075 Coimbra, Portugal; sirivpaulo@gmail.com (S.P.); josepedromartinho@gmail.com (J.P.M.); m.mferreira@netcabo.pt (M.M.F.)

**Keywords:** endodontics, root canal sealer, root canal filling materials, cell death, biocompatibility, systematic review

## Abstract

(1) Aim: To perform a systematic review of the literature on the biocompatibility of root canal sealers that encompasses the various types of sealers that are commercially available as well as both in vitro and in vivo evidence. (2) Methods: This systematic review has been registered in PROSPERO (ID 140445) and was carried out according to PRISMA guidelines using the following databases: PubMed, Cochrane Library, ClinicalTrials.gov, Science Direct, and Web of Science Core Collection. Studies published between 2000 and 11 June 2019 that evaluated cytotoxicity (cell viability/proliferation) and biocompatibility (tissue response) of root canal sealers were included. (3) Results: From a total of 1249 studies, 73 in vitro and 21 in vivo studies were included. In general, studies suggest that root canal sealers elicit mild to severe toxic effects and that several factors may influence biocompatibility, e.g., material setting condition and time, material concentration, and type of exposure. Bioactive endodontic sealers seem to exhibit a lower toxic potential in vitro. (4) Conclusions: The available evidence shows that root canal sealers exhibit variable toxic potential at the cellular and tissue level. However, the methodological heterogeneity among studies included in this systematic review and the somewhat conflicting results do not allow a conclusion on which type of sealer presents higher biocompatibility. Further research is crucial to achieve a better understanding of the biological effects of root canal sealers.

## 1. Introduction

Root canal therapy encompasses the sequence of procedures with the aim of treating the infected canal of a tooth, thus resulting in the resolution of the infectious process and in the prevention of microbial invasion in the intervened tooth [1].

The usage of endodontic sealers to perform root canal fillings in obturation procedures is an established mainstay in endodontics and plays a key role in the success of the treatment [2]. Therefore, these materials should exhibit a set of characteristics that allow successful root canal filling with resolution of periapical inflammatory and/or infectious processes and prevent further microbial contamination [2]. In this context, Grossman previously listed the properties of an ideal sealer: (a) exhibits tackiness when mixed to provide good adhesion to the canal wall, (b) establishes a hermetic seal, (c) is radiopaque, so that it can be observed through radiographic observation, (d) is a very fine powder that can be easily mixed with liquid, (e) does not shrink on setting, (f) does not stain tooth structure, (g) is bacteriostatic (or at least does not promote bacterial growth), (h) displays a slow setting, (i) is insoluble in host tissue fluids, (j) is biocompatible, i.e., without irritant potential to periradicular tissue, and (k) is soluble in common solvent, allowing for removal when necessary [3].

Over the years, scientific and technological advances have allowed the improvement of the equipment and materials used in several areas, particularly in endodontics, thus providing better results [4]. However, no sealer has yet fulfilled the entire set of Grossman’s criteria [2].

In fact, a number of materials have been developed, which may be categorized into the following classes according to their chemical composition and structure: zinc oxide-eugenol-based, resin-based, glass ionomer-based, silicone-based, calcium hydroxide-based, and bioactive endodontic sealers. The physical, chemical, and biological properties have been previously reviewed [5,6,7].

As mentioned above, biocompatibility is one of the main properties of root canal sealers, as these materials come into direct contact with periradicular tissues [2]. This biocompatibility corresponds to the ability to achieve an appropriate host response in a specific application; i.e., when in contact with the tissue, it fails to trigger an adverse reaction [6,8,9]. However, all sealers tend to exhibit a certain degree of toxicity especially when in a freshly mixed state, even though the toxicity tends to decrease with setting [2,10]. Therefore, the extrusion of sealer into periradicular tissues should be avoided [2].

Most studies evaluate such biocompatibility through an in vitro assessment of cytotoxicity with cell models [11]. Furthermore, multiple in vivo studies that assess tissue response have also been published. However, the multiplicity of methods and conditions that have been tested in previous studies makes it difficult to get an overview of the subject as well as its interpretation. This integration of concepts and results may be achieved through the systematic review of the literature.

In this context, we aimed to perform a systematic review of the literature on the in vitro cytotoxicity (as a measure of direct cellular toxicity) and in vivo biocompatibility (as inflammatory tissue reaction) of root canal sealers. As previous systematic reviews have focused on the superiority of calcium silicate-based sealers [12,13,14], here we aimed to include all types of sealers and both in vitro and in vivo studies in order to present a more complete perspective on the biocompatibility of endodontic sealers as well as to compare the results and understand how the evidence correlates between both types of study. Furthermore, we also aimed at understanding how the material set condition and concentration and the type and time of exposure influence the cytotoxicity and biocompatibility of these materials. Clinically, this systematic review aims to provide an integrated perspective on the biocompatibility of root canal sealers and the main factors that may influence endodontic treatment outcome and success from a biocompatibility standpoint.

## 2. Methods

This systematic review was carried out according to the Preferred Reporting Items for Systematic Reviews and Meta-Analyses (PRISMA) guidelines [15] and was registered in PROSPERO with the ID 140445. Considering the non-clinical nature of this systematic review, the PICO (Population, Intervention, Comparison, and Outcome) research question was adapted from the PICO framework [16] (Table 1) and was formulated as follows: How do root canal sealers (individually or by type) perform in terms of cytotoxicity and biocompatibility in experimental cell and animal models?

### 2.1. Search Strategy and Study Selection

The electronic search was performed in several databases, specifically Medline via PubMed, Cochrane Library, ClinicalTrials.gov, Science Direct, Web of Science Core Collection, and EMBASE. The last search was performed on 11 June 2019, and a date limit was applied to include studies published since the introduction of AH Plus^TM^ (Dentsply DeTrey Gmbh, Konstanz, Germany), i.e., since 2000, as this has been the most studied sealer in the last two decades [17]. Furthermore, the following language filters were applied: English, Portuguese, and Spanish. The search equations used for each electronic database are detailed in Table A1 (Appendix B).

Articles were initially screened based on the title and abstract according to the scope (i.e., articles that do not report the cytotoxicity and/or biocompatibility of endodontic sealers for root canal filling) and publication type (i.e., reviews, comments, letters, or abstracts). Furthermore, a hand search of the reference lists of relevant studies was also performed. Reference management was performed with Mendeley^©^ v1.19.4 (Mendeley Ltd, London, United Kingdom).

In the eligibility assessment phase, this systematic review was split into two main sections based on the population and the outcomes: (a) one referring exclusively to in vitro models of cytotoxicity assessment and (b) another referring exclusively to in vivo animal models of biocompatibility assessment. Two independent reviewers critically assessed the eligibility of studies for inclusion, collected the data, and assessed the risk of bias. A third reviewer was consulted in case of uncertainty or discrepancies, and a decision was made by consensus.

For the in vitro section, in vitro studies that evaluated the cytotoxicity, by assessing cell viability/proliferation of root canal sealers were included, and the following exclusion criteria were considered: (i) studies whose cytotoxicity assessment method is not clear or incompletely described or that do not evaluate or only evaluate the cytotoxicity of endodontic sealers for root canal filling qualitatively; (ii) studies that do not evaluate cytotoxicity through methods specific for cell viability/proliferation evaluation; (iii) studies that only report other biological properties (e.g., antimicrobial effect), physicochemical properties (e.g., bond strength, radiopacity, pH, solubility, setting time, working time, dimensional change, flow, or calcium release) or clinical outcomes (e.g., apical leakage or adaptation, sealing ability); (iv) studies that report the cytotoxic effects of experimental sealers that are not commercially available, modified commercially available root canal sealers, modified sealer components, or dental materials used as pulp-capping materials and others (e.g., adhesive systems); and (v) studies other than in vitro, e.g., in vivo or in silico.

For the in vivo section, in vivo animal studies that evaluated the biocompatibility of root canal sealers through the assessment of tissue reaction after subcutaneous, intraosseous, alveolar socket, or root canal implantation were included. For this section, the following exclusion criteria were considered: (i) studies that do not report the biocompatibility of endodontic sealers for root canal filling according to the methods described in the inclusion criteria; (ii) studies that only report other biological properties or clinical outcomes; (iii) studies that report the biocompatibility of experimental sealers that are not commercially available, modified commercially available root canal sealers or dental materials used as pulp capping materials; and (iv) studies other than in vivo, e.g., in vitro.

Studies with missing data were excluded, particularly regarding the method used for cytotoxicity or biocompatibility assessment, the material setting condition, the sealer extraction time and/or extract concentration, and cell incubation time for in vitro studies.

### 2.2. Data Collection

The following descriptive and quantitative information was extracted from each of the eligible studies for both sections, i.e., in vitro and in vivo: authors and year of publication, tested sealer(s) and controls, sample size, sealer material condition (i.e., fresh or set), the setting time if set materials were used, method of sealer preparation (i.e., if in accordance to manufacturer’s instructions), results, and main conclusions. Relative to in vitro studies, the following information was also extracted: method (i.e., direct or indirect contact with sealer specimens or extracts), extraction time and extract concentration if extracts were obtained, cell model, exposure time, and cell viability/proliferation assay. In regard to in vivo studies, the following information was also extracted: method of biocompatibility assessment (i.e., subcutaneous, alveolar, intraosseous, or root canal implantation), teeth used for root canal filling if this method was used, animal model, exposure time, and method of histologic analysis (including staining method and outcomes measured).

### 2.3. Risk of Bias

The methodological quality of eligible studies was checked by assessing the risk of bias of individual studies. For in vitro studies, the guidelines for reporting of preclinical studies on dental materials by Faggion Jr. [18] were followed, consisting of several items that were based on the Consolidated Standards of Reporting Trials (CONSORT) guidelines for reporting randomized clinical trials. For in vivo studies, the Systematic Review Centre for Laboratory Animal Experimentation (SYRCLE) risk of bias tool [19] was used, which represents an adapted version of the Cochrane’s risk of bias tool.

## 3. Results

The full process of article retrieving, screening, and eligibility assessment is presented in Figure 1. The initial search retrieved a total of 1444 studies, from which 195 were excluded after the removal of duplicates. A total of 1249 studies were screened based on the title and abstract, from which 1068 were excluded, resulting in 181 full-text studies that were considered potentially eligible for inclusion, including 146 in vitro studies, 32 in vivo studies, and three studies with both in vitro and in vivo testing. A total of 102 studies (82 in vitro, 18 in vivo and two both in vitro and in vivo) was excluded because they did not meet the inclusion criteria. Studies that did not specify the material condition, i.e., freshly mixed or set, were excluded. After reviewing full texts, seven in vitro, seven in vivo and one both in vitro and in vivo studies were added to the analysis by hand searching. Finally, 71 in vitro, 21 in vivo and two both in vitro and in vivo studies were included in this review. Two studies with both in vitro and in vivo methodologies [20,21] were included only for the in vitro data, as the in vivo methodology did not meet the inclusion criteria. A list of the various endodontic materials studied in eligible studies and respective manufacturers is included as Appendix A.

As can be seen, the most studied sealers in vitro were: AH 26^®^ (Dentsply DeTrey Gmbh, Konstanz, Germany), AH Plus^TM^, EndoREZ^®^ (Ultradent Products Inc., South Jordan, UT, USA), Endosequence BC^TM^ (Brasseler, Savannah, GA, USA), Epiphany^®^ (Pentron, Wallingford, CT, USA), MTA Fillapex^®^ (Angelus, Londrina, Brazil), Kerr’s Pulp Canal Sealer^TM^ (PCS; Kerr, Romulus, MI, USA), and Sealapex^TM^ (Kerr, Romulus, MI, USA). Regarding in vivo studies, AH Plus^TM^, EndoREZ^®^ and Epiphany^®^ were the most studied.

### 3.1. In Vitro Cytotoxicity

The methodological characteristics of the included in vitro studies are presented in Table A2 (Appendix C). Of the 73 studies, 18 used a direct contact testing method with sealers prepared either as fresh sample, disc, layer or cylindrical specimens [17,22,23,24,25,26,27,28,29,30,31,32,33,34,35,36,37,38], while others used sealer specimens on inserts [39,40,41,42,43,44] or root models [45,46,47,48]. In terms of the material setting condition, 23 studies evaluated root canal sealers in fresh or freshly mixed state, 15 in set condition with 24 h incubation, 15 in both freshly mixed and set conditions, and 20 in a set condition with other or multiple times of incubation.

Concerning the cell models used for cell viability assessment (Table A2), several studies used cultures of human cells, namely: dental follicle-derived mesenchymal stem cells [23], tooth germ-derived stem cells [40], bone marrow-derived mesenchymal stem cells [49], gingival fibroblasts [31,50,51,52,53,54,55,56], dental pulp stem cells [57], osteoblasts [23,41,46,58,59,60,61], periodontal ligament cells [26,28,39,45,48,62,63,64,65,66], human osteoblast-like cells (MG63) [38,42,67,68], cervical carcinoma cells or human cervical carcinoma cells (HeLa) cells [33,37] and THP-1 human monocytic cells [36]. Other cell lines were also used, e.g., L929 mouse fibroblasts, mouse osteoblast-like cells (MC3T3-E1), RAW 264.7 mouse macrophages, Chinese hamster fibroblasts (V79), rat osteosarcoma (ROS) 17/12.8 cells, Balb/c fibroblasts, and rat clonal dental pulp cells (RPC-C2A).

Regarding the type of cell viability assay (Table A2), most of the studies used assays that measure metabolic activity, specifically: 39 studies used the 3-[4,5-dimethylthiazol-2-yl]-2,5- diphenyltetrazolium bromide (MTT) assay, two used the 2,3-bis-(2-methoxy-4-nitro-5-sulfophenyl)-2H-tetrazolium-5-carboxanilide (XTT) assay, four used the Alamar blue^®^ assay, three used the Cell Counting Kit-8 (CCK-8/WST-8) assay, two used the Water Soluble Tetrazolium Salt-1 (WST-1) assay, and one used the 3-(4,5-dimethylthiazol-2-yl)-5-(3-carboxymethoxyphenyl)-2-(4-sulfophenyl)-2H-tetrazolium (MTS) assay. Other methods included the Trypan blue dye exclusion assay, the Neutral Red uptake assay, and the Sulforhodamine B assay, among others. In addition, five studies used multiple methods to assess cell viability.

#### 3.1.1. Cytotoxicity of Root Canal Sealers

In general, the tested root canal sealers exhibited cytotoxicity (Table 2). The most studied sealer was the epoxy resin-based sealer AH Plus^TM^, which was reported as cytotoxic in most of the studies in which it was tested. However, one study [47] reported it as noncytotoxic, one [55] reported a cytotoxic effect only in the early phase, two [61,66] reported it as cytotoxic only in fresh conditions, and one [69] reported it as cytotoxic when eluted in dimethyl sulfoxide (DMSO) but noncytotoxic when eluted in sodium chloride.

Similarly, PCS showed cytotoxicity in all the studies, except for one [35]. In addition, the formaldehyde-releasing epoxy resin-based sealer AH 26^®^ and the zinc oxide-eugenol-based sealer N2^®^ (Indrag-Agsa, Losone, Switzerland) showed cytotoxic effects in all the studies.

Regarding bioactive sealers, several studies reported no cytotoxic effect for BioRoot^TM^ RCS (Septodont, Saint-Maur-des-Fossés, France) [24,62,63], mineral trioxide aggregate (MTA) sealers [21,26,70,71], iRoot^®^ (Innovative BioCeramix Inc., Vancouver, Canada) sealers [68,70,72], and Endosequence BC^TM^ [51]. However, a cytotoxic effect of bioactive sealers was also reported in comparison with other materials, either similar—compared to epoxy resin-based [23,51,64,73,74] or calcium hydroxide-based [26] sealers—or lower compared to zinc oxide-eugenol-based [39,41,45,46,66,74,75], epoxy resin-based [23,39,40,49,52,61,66,73,74,75,76], methacrylate resin-based [39,41], or other materials [67]. Some studies reported a higher cytotoxic effect of MTA Fillapex^®^ compared with epoxy resin-based sealers in set condition [23,40,51,57,61,66,71]. Although one study [51] showed no cytotoxic effect by Endosequence BC^TM^, one study [49] showed lower cytotoxicity than the epoxy resin-based AH Plus^TM^ and similar to BioRoot^TM^ RCS, and one study [43] showed a higher cytotoxicity compared to AH Plus^TM^ in set material conditions but lower than PCS.

In respect to other materials, no cytotoxic effect was reported for the silicone-based sealers GuttaFlow^®^2 (Roeko/Coltène/Whaledent, Langenau, Germany) [52,77] and GuttaFlow^®^ Bioseal (Roeko/Coltène/Whaledent, Langenau, Germany) [64], even though a cytotoxic effect has also been shown for GuttaFlow^®^2 [64]. In addition, Mendes et al. [35] reported no cytotoxic effect for the zinc oxide-eugenol-based sealer Endofill (Produits Dentaires, Vevey Switzerland), although other studies showed cytotoxicity [20,41,78]. In addition, the cytotoxicity of urethane dimethacrylate (UDMA) and polymethyl methacrylate (PMMA) was also reported by Lee et al. [79] and Pinna et al. [29], respectively. Furthermore, one study [47] showed no cytotoxic effect for the calcium hydroxide-based sealer Sealapex^TM^ in set material condition. However, other studies showed lower [22,80,81], similar [26], and higher [26,47] cytotoxicity compared to other sealers. One study [31] showed opposing cytotoxic potential according to the setting condition, as Sealapex^TM^ exhibited lower cell toxicity in fresh material conditions (1 h after mixing) compared to the set material conditions (24 h after preparation).

Generally, the results from the included studies suggested that bioactive sealers may exhibit lower cytotoxic potential compared to other types of root canal sealer.

**Table 2 materials-12-04113-t002:** Summary of parameters and results collected from included in vitro studies, ordered by publication date (from most recent).

Year	Study	Groups	Sealer–Cell Contact	Extraction Time	Extract Concentration	Cell Exposure Time	Cytotoxic Potential
2019	Lee et al. [76]	AH Plus^TM^, Mineral Trioxide Aggregate (MTA) Fillapex^®^, Endosequence BioCeramic (BC) ^TM^, Medium (control)	Indirect (extract)	7 d	1, 1:5, 1:10, 1:50, 1:100	1 d	Endosequence BC^TM^ < MTA Fillapex^®^ < AH Plus^TM^
	Jeanneau et al. [62]	BioRoot^TM^ Root Canal Sealer (RCS), Kerr’s Pulp Canal Sealer (PCS), Medium (control)	Indirect (extract)	1 d	0.2 mg/mL	3 d, 6 d, 9 d	BioRoot^TM^ RCS (nontoxic) < PCS
	Giacomino et al. [74]	Roth´s Sealer, AH Plus^TM^, Endosequence BC^TM^, ProRoot^®^ Endodontic Sealer (ES), No cells (control), Medium (control)	Indirect (extract)	3 d	Several dilutions	7 d	Endosequence BC^TM^ < ProRoot^®^ ES < Roth’s, AH Plus^TM^
	Jung et al. [66]	MTA Fillapex^®^, BioRoot^TM^ RCS, AH Plus^TM^, PCS, Medium (control)	Indirect (extract)	1 d	1:1, 1:2, 1:10	1 d, 7 d, 14 d, 21 d	BioRoot^TM^ RCS < AH Plus^TM^ (toxic only in fresh) < MTA Fillapex^®^, PCS (toxic as fresh or set)
2018	Vouzara et al. [73]	SimpliSeal^®^, MTA Fillapex^®^, BioRoot^TM^ RCS, Medium (control)	Indirect (extract)	1 d, 1 w	1:1, 1:2	1 d, 3 d	BioRoot^TM^ RCS < MTA Fillapex^®^, SimpliSeal^®^
	Alsubait et al. [49]	AH Plus Jet^®^, Endosequence BC^TM^, BioRoot RCS^TM^, Medium (control)	Indirect (extract)	1 d	1:2, 1:8, 1:32	1 d, 3 d, 7 d	Endosequence BC^TM^, BioRoot^TM^ RCS < AH Plus Jet^®^
	Jung et al. [61]	AH Plus^TM^, PCS, MTA Fillapex^®^, BioRoot^TM^ RCS, Medium (control)	Indirect (extract)	1 d	1:1, 1:2, 1:10	1 d, 7 d, 14 d, 21 d	BioRoot^TM^ RCS < AH Plus^TM^ (toxic only in fresh) < MTA Fillapex^®^, PCS (toxic as fresh or set)
	Szczurko et al. [39]	AH Plus Jet^®^, Apexit^®^ Plus, MTA Fillapex^®^, GuttaFlow^®^, MetaSEAL^TM^ Soft, Tubli-Seal^TM^, Untreated (control)	Indirect (sealer on insert)	-	-	1 d	Fresh: GuttaFlow^®^ < Apexit^®^ Plus, MTA Fillapex^®^ < AH Plus Jet^®^, Tubli-Seal^TM^ < MetaSEAL^TM^ (did not compare fresh vs set conditions)
	Troiano et al. [38]	AH Plus^TM^, Sicura Seal, TopSeal^®^, Medium (control)	Direct and indirect (extract)	Several time points	n/s	1 d, 2 d, 3 d, 7 d (direct) and 1 d (indirect)	All cytotoxic (no major differences among sealers). Direct cytotoxicity decreased over time.
2017	Arun et al. [22]	Tubli-Seal^TM^, AH Plus^TM^, Sealapex^TM^, EndoREZ^®^, Medium (control) (groups with pachymic acid)	Direct	-	-	1 d	Sealapex^TM^ < AH Plus^TM^ < Tubli-Seal^TM^ < EndoREZ^®^
	Collado-González et al. [63]	BioRoot^TM^ RCS, Endoseal^®^, Nano-ceramic Sealer (NCS), Medium (control)	Indirect (extract)	1 d	1:1, 1:2, 1:4	1 d, 2 d, 3 d	BioRoot^TM^ RCS (biocompatible) < NCS < Endoseal^®^
	Collado-González et al. [64]	GuttaFlow^®^ Bioseal, GuttaFlow^®^2, MTA Fillapex^®^, AH Plus^TM^, Medium (control)	Indirect (extract)	1 d	Undiluted, 1:2, 1:4	1 d, 2 d, 3 d, 7 d	GuttaFlow^®^ Bioseal (nontoxic) < GuttaFlow^®^2, AH Plus^TM^, MTA Fillapex^®^
	Cintra et al. [21]	MTA High plasticity, MTA Angelus^®^, Medium (control)	Indirect (extract)	3 d	1:50	6 h, 1 d, 2 d, 3 d	MTA High Plasticity (nontoxic) < MTA Angelus^®^
	Zhu et al. [72]	iRoot^®^ Sealing Paste (SP), MTA, Medium (control)	Indirect (extract)	1 d	Undiluted	1 d, 2 d	iRoot^®^ SP, MTA (nontoxic)
	Cintra et al. [20]	Sealer Plus, AH Plus^TM^, Endofill, SimpliSeal^®^, Medium (control)	Indirect (extract)	3 d	Undiluted, 1:2, 1:4	6 h, 1 d, 2 d, 3 d	Sealer Plus < SimpliSeal^®^ < AH Plus^TM^, Endofill
	Lv et al. [70]	iRoot^®^ Fast Setting (FS), iRoot^®^ Bioceramic Putty (BP) Plus, ProRoot^®^ MTA, Medium (control)	Indirect (extract)	3 d	Undiluted, 1:2, 1:4	1 d, 2 d, 3 d	iRoot^®^ FS, iRoot^®^ BP Plus, ProRoot^®^ MTA (nontoxic)
	Victoria-Escandell et al. [57]	MTA Angelus^®^, MTA Fillapex^®^, AH Plus^TM^, Medium (control)	Indirect (extract)	1 d, 2 d, 7 d, 15 d, 30 d	1:2	1 d	MTA Angelus^®^ (less toxicity) < AH Plus^TM^ < MTA Fillapex^®^
2016	Suciu et al. [23]	MTA Fillapex^®^, AH Plus^TM^, Acroseal, Plastic surface (control)	Direct	-	-	2 d, 5 d, 9 d, 14 d	hOCs (human osteoblastic cells): Acroseal, MTA Fillapex^®^ < AH Plus^TM^. DF-MSCs (dental follicle-derived adult mesenchymal stem cells): Acroseal < AH Plus^TM^ < MTA Fillapex^®^
2015	Camps et al. [45]	BioRoot^TM^ RCS, PCS, Medium (control)	Indirect (extract from root model)	1 d	Undiluted	2 d, 5 d, 7 d	BioRoot^TM^ RCS < PCS
	Dimitrova-Nakov et al. [24]	BioRoot^TM^ RCS, PCS, Untreated cells (controls)	Direct	-	-	7 d, 10 d	BioRoot^TM^ RCS (nontoxic) < PCS
	Konjhodzic-Prcic et al. [50]	GuttaFlow^®^, AH Plus^TM^, Apexit^®^, EndoREZ^®^, Control (n/s)	Indirect (extract)	1 d	Undiluted	1 d	All slightly cytotoxic
	Konjhodzic-Prcic et al. [82]	GuttaFlow^®^, AH Plus^TM^, Apexit^®^, EndoREZ^®^, Control (n/s)	Indirect (extract)	1 d	Undiluted	1 d	Apexit^®^, GuttaFlow^®^, AH Plus^TM^ < EndoREZ^®^
	Zhou et al. [51]	Endosequence BC^TM^, MTA Fillapex^®^, Medium (control)	Indirect (extract)	Fresh: 1 d. Set: 1 d, 1 w, 2 w, 3 w, 4 w	1:2, 1:8, 1:32, 1:128	3 d	Endosequence BC^TM^ (nontoxic). Fresh: MTA Fillapex^®^ < AH Plus^TM^. Set: AH Plus^TM^ < MTA Fillapex^®^
	Silva et al. [77]	GuttaFlow^®^2, AH Plus^TM^, Medium (control)	Indirect (extract)	1 d to 3 d	Undiluted	1 d	GuttaFlow^®^2 (nontoxic) < AH Plus^TM^
	Parirokh et al. [56]	Duraflur^®^, AH Plus^TM^, AH 26^®^, Medium (control)	Indirect (extract)	1 d	1/2, 1/4, 1/8	1 d	AH Plus^TM^ < Duraflur^®^ < AH 26^®^ (concentration-dependent)
2014	Jiang et al. [67]	iRoot^®^ BP Plus, iRoot^®^ FS, ProRoot^®^ MTA, SuperEBA^TM^, Medium (control)	Indirect (extract)	1 d, 3 d, 7 d, 14 d	100%, 50%, 25%	1 d	iRoot^®^ BP Plus, iRoot^®^ FS, ProRoot^®^ MTA < SuperEBA^TM^
	Cotti et al. [25]	RealSeal XT, AH Plus Jet^®^, Untreated (control)	Direct	-	-	1 h, 1 d, 2 d, 3 d	RealSeal XT < AH Plus Jet^®^
	Chang et al. [26]	Sealapex^TM^, Apatite Root Sealer, MTA Fillapex^®^, iRoot^®^ SP, Medium with and without osteogenic supplementation (O.S.) (control)	Direct	-	-	3 d, 7 d, 14 d	MTA Fillapex^®^ (nontoxic) < Sealapex^TM^, Apatite Root Sealer, iRoot^®^ SP
	Mandal et al. [52]	GuttaFlow^®^2, ProRoot^®^ MTA, AH Plus^TM^, RealSeal^TM^, Medium (control)	Indirect (extract)	1 d, 3 d	0.5, 1, 1.5 cm^2^/mL	1 d	GuttaFlow^®^2 (nontoxic as fresh), ProRoot^®^ MTA < AH Plus^TM^, RealSeal^TM^
	Camargo et al. [83]	AH Plus^TM^, EndoREZ^®^, RoekoSeal, Medium (control)	Indirect (extract)	1 d	1:1, 1:2, 1:4, 1:8, 1:16, 1:32	1 d	RoekoSeal < AH Plus^TM^ < EndoREZ^®^
2013	Güven et al. [40]	MTA Fillapex^®^, iRoot^®^ SP, AH Plus Jet^®^, Control (n/s)	Indirect (sealer on insert)	-	-	1d, 3d, 7d, 14d	iRoot^®^ SP < AH Plus^TM^ < MTA Fillapex^®^
	Kim et al. [84]	AH Plus^TM^ (in the presence or absence of pachymic acid and NAC)	Indirect (extract)	1 d	30%	1d	AH Plus^TM^ was cytotoxic
2012	De-Deus et al. [46]	iRoot^®^ BP Plus, ProRoot^®^ MTA, Medium (negative control), zinc oxide-eugenol (ZOE) cement (positive control)	Indirect (extract from root model)	1 d, 2 d	Undiluted	1d	ProRoot^®^ MTA (nontoxic) < iRoot BP Plus < ZOE
	Bin et al. [71]	MTA Angelus^®^, MTA Fillapex^®^, AH Plus^TM^, Untreated (control)	Indirect (extract)	1 d	1:1, 1:2, 1:4, 1:8, 1:16, 1:32	1d	MTA Angelus^®^ (nontoxic) < AH Plus^TM^ < MTA Fillapex^®^
	Scelza et al. [53]	RealSeal Self-Etch (SE) ^TM^, AH Plus^TM^, GuttaFlow^®^, Sealapex^TM^, Roth 801, ThermaSeal^®^ Plus, Medium (control)	Indirect (extract)	1 d, 7 d, 14 d, 21 d, 28 d	Undiluted	1d	GuttaFlow^®^ < AH Plus^TM^ < ThermaSeal^®^ Plus < Roth 801 < RealSeal^TM^ < Sealapex^TM^
	Salles et al. [41]	MTA Fillapex^®^, Epiphany^®^ SE, Endofill, Untreated (control)	Indirect (sealer on insert)	-	-	1d, 2d, 3d, 7d	MTA Fillapex^®^ (toxic only for 3d) < Epiphany^®^ SE, Endofill
	Landuyt et al. [54]	AH Plus Jet^®^, EndoREZ^®^, RealSeal^TM^, Calcicur (control), Medium (negative control), 1% Triton X-100 (positive control)	Indirect (extract)	1 d	1:1, 1:3, 1:10, 1:30, 1:100, 1:300	1d	EndoREZ^®^ < RealSeal^TM^ < AH Plus Jet^®^
	Shon et al. [42]	CAPSEAL I and II, Apatite Root Sealer type I and III, PCS Extended Working Time (EWT), Medium (control)	Indirect (sealer on insert)	-	-	18h, 1d, 3d, 7d, 14d	CAPSEAL < Apatite Root Sealer < PCS EWT (cytotoxicity increased with time for Apatite Root Sealers and PCS EWT)
2011	Mukhtar-Fayyad [85]	BioAggregate^®^, iRoot^®^ SP, Medium (control)	Indirect (extract)	5 d	Undiluted, 1:2, 1:10, 1:50, 1:100	1d, 3d, 7d	iRoot^®^ SP < BioAggregate^®^ (concentration-dependent)
	Zoufan et al. [75]	GuttaFlow^®^, Endosequence BC^TM^, AH Plus Jet^®^, TubliSeal Xpress^TM^, Untreated (control)	Indirect (extract)	1 d, 3 d	Eluates (300, 600 and 1000 μL)	1 d	GuttaFlow^®^, Endosequence BC^TM^ less toxic. F^1^: Tubli-Seal Xpress^TM^ < AH Plus^TM^. Set^1^: AH Plus^TM^ < Tubli-Seal Xpress^TM^
	Loushine et al. [43]	Endosequence BC^TM^, AH Plus^TM^, PCS EWT (positive control), Teflon (negative control)	Indirect (sealer on insert)	-	-	1 d/week (for 6 weeks)	AH Plus^TM^ < Endosequence BC^TM^ < PCS
	Brackett et al. [36]	AH Plus Jet^®^, PCS, ProRoot^®^ MTA, Experimental calcium-silicate sealer, Teflon (control)	Direct	-	-	3 d	ProRoot^®^ MTA, Experimental sealer < AH Plus Jet^®^ < PCS
2010	Yu et al. [86]	AH 26^®^, Control (n/s)	Indirect (extract)	1 d, 3 d, 5 d, 7 d	30%	1 d, 2 d	AH Plus^TM^ was cytotoxic (extraction time-dependent)
	Zhang et al. [68]	iRoot^®^ SP, AH Plus^TM^, Medium (control)	Indirect (extract)	1 d	1:1, 1:2, 1:4	1 d	iRoot^®^ SP (nontoxic) < AH Plus
	Huang et al. [58]	AH 26^®^, Canals, N2^®^, Untreated (control)	Indirect (extract)	1 d	1:2, 1:4, 1:8	1 d	Canals < N2^®^ < AH 26^®^ (concentration-dependent)
	Bryan et al. [44]	Experimental sealer (calcium silicate-based), AH Plus^TM^, PCS, Teflon (negative control)	Indirect (sealer on insert)	-	-	3 d/week (for 5 weeks)	Experimental sealer < AH Plus^TM^ < PCS (concentration-dependent)
2009	Ames et al. [27]	EndoREZ^®^, RealSeal^TM^, MetaSEAL^TM^, RealSeal SE^TM^, PCS (positive control), Teflon (negative control)	Direct	-	-	3 d/week (for 5 weeks)	RealSeal SE^TM^, MetaSEAL^TM^ (both ↓ with time) < EndoREZ^®^, RealSeal^TM^, PCS
	Donadio et al. [87]	Activ Gutta-Percha (GP) ^TM^, RealSeal^TM^, AH 26^®^, Kerr Sealer, Untreated (control)	Indirect (extract)	1 d, 3 d	Eluates (200, 400, 800 and 1200 μL)	1 d	Fresh ^1^: Kerr < RealSeal^TM^, Activ GP^TM^ < AH 26^®^ Set ^1^: AH 26^®^, Kerr < Activ GP^TM^ < RealSeal^TM^
	Gambarini et al. [88]	Epiphany^®^ SE, Epiphany^®^, PCS, Untreated (control)	Indirect (extract)	1 d	Undiluted	1 d	Epiphany^®^, Epiphany^®^ SE, PCS
	Camargo et al. [89]	AH Plus^TM^, Epiphany^®^, Acroseal, Castor Oil Polymer sealer, Untreated (control)	Indirect (extract)	1 d	1:1, 1:2, 1:4, 1:8, 1:16, 1:32	1 d	Castor Oil Polymer << AH Plus^TM^, Epiphany^®^ < Acroseal
	Huang et al. [59]	AH 26^®^, Canals, N2^®^, Untreated (control)	Indirect (extract)	1 d	1:2, 1:4, 1:8	2 d	Canals < AH 26^®^, N2^®^ (concentration-dependent)
2008	Heitman et al. [28]	Epiphany^®^, Untreated (control)	Direct	-	25, 50, 100, 200, 400, 800 μg/mL	1 d, 3 d, 7 d	Epiphany^®^ was cytotoxic (concentration- and exposure time-dependent)
	Valois and Azevedo [78]	AH Plus^TM^, Endofill, Sealer 26, Medium from empty molds (control)	Indirect (extract)	1 d	20%, 10%, 5%	1 d	All cytotoxic (concentration-dependent)
	Pinna et al. [29]	MetaSEAL^TM^, AH Plus Jet^®^, PCS, polymethyl methacrylate (PMMA, positive control), Teflon (negative control)	Direct	-	-	3 d/week (for 5 weeks)	AH Plus Jet^®^, PMMA < MetaSEAL^TM^ < PCS (time-dependent, except for PCS)
	Huang et al. [60]	AH 26^®^, Canals, N2^®^, Untreated (control)	Indirect (extract)	1 d	1:2, 1:4, 1:8	2 d	Canals < AH 26^®^ < N2^®^ (concentration-dependent)
	Lodienė et al. [30]	AH Plus^TM^, EndoREZ^®^, RoekoSeal Automix, Epiphany^®^, Medium (control)	Direct and indirect (extract)	1 d (set)	Undiluted	2 h	EndoREZ^®^ < AH Plus^TM^, RoekoSeal < Epiphany^®^
2007	Lee et al. [80]	N2^®^, Sealapex^TM^, AH 26^®^, Control (n/s)	Indirect (extract)	1 d	Dilution factor: 10–80	1 d	Sealapex^TM^ < AH 26^®^ < N2^®^ (concentration-dependent)
	Lee et al. [79]	AH 26^®^, urethane dimethacrylate (UDMA), Control (n/s)	Indirect (extract)	1 d	5 mg/mL and dilutions	1 d	Cytotoxicity was concentration-dependent (prevented by NAC)
	Lee et al. [81]	N2^®^, Sealapex^TM^, AH 26^®^, Control (n/s)	Indirect (extract)	1 d	Dilution factors: 6–18, 1–7, 5–100	1 d	Sealapex^TM^ < N2^®^ < AH 26^®^ (concentration-dependent)
	Merdad et al. [37]	Epiphany^®^, AH Plus^TM^, Filters with cells and no sealer, and filters with no cells and with sealer (controls)	Direct and indirect (specimens)	-	-	2 h	Epiphany^®^ < AH Plus^TM^
2006	Key et al. [31]	Epiphany^®^, Resilon, GP, Grossman, Thermaseal^®^, Sealapex^TM^. Isotonic saline and 10% formaldehyde (controls)	Direct	-	-	1 h, 1 d	F^1^: Sealapex^TM^ < others. S^1^: Thermaseal^®^, Epiphany^®^ < others.
	Bouillaguet et al. [32]	AH Plus^TM^, Epiphany^®^, GuttaFlow^®^, Teflon (control)	Direct	-	-	1 d, 3 d	GuttaFlow^®^ < AH Plus^TM^ < Epiphany^®^ (exposure time-dependent)
2005	Miletic et al. [33]	Roekoseal Automix, AH Plus^TM^, Control (n/s)	Direct	-	-	5 d	RoekoSeal < AH Plus^TM^ (setting time-dependent for AH Plus^TM^)
2004	Al-Awadhi et al. [90]	Sealapex^TM^, PCS, Roekoseal Automix, Medium (control)	Indirect (extract)	1 d	190 mm^2^/1 mL, 50 or 300 μL (b, ED50)	(a) 1 d (b) 1 d, 3 d	(a) RoekoSeal, Sealapex^TM^ < PCS (b) RoekoSeal < PCS, Sealapex^TM^
	Bouillaguet et al. [34]	PCS, RoekoSeal, TopSeal^®^, EndoREZ^®^, Teflon (control)	Direct	-	-	1 d 1 d, 7 d	RoekoSeal < PCS, TopSeal^®^, EndoREZ^®^ (both fresh and set)
2003	Camps and About [47]	AH Plus^TM^, Cortisomol^TM^, Sealapex^TM^, Medium (control)	Indirect (normal extracts and from root model)	1 d, 2 d, 30 d	Undiluted	1 d	(a) AH Plus^TM^ < Cortisomol^TM^ < Sealapex^TM^ (b) Sealapex^TM^ < AH Plus^TM^ < Cortisomol^TM^
	Mendes et al. [35]	PCS, Endofill, Medium (control)	Direct	-	-	2 h, 1 d, 2 d	PCS, Endofill (nontoxic)
2002	Schwarze et al. [48]	AH Plus^TM^, Apexit^®^, Endométhasone, Ketac^TM^ Endo, N2^®^, RoekoSeal, Gutta-percha, Medium (control)	Indirect (extract)	24 h, 1–52 w	Undiluted	1 d	Pronounced cytotoxicity only by N2^®^
	Huang et al. [91]	AH 26^®^, AH Plus^TM^, Medium and dimethyl sulfoxide (DMSO) as controls	Indirect (extract)	1 d	0.10, 0.08, 0.04, 0.02, 0.01 mg/mL	1 d	Both cytotoxic (concentration-dependent)
	Schwarze et al. [65]	N2^®^, Endométhasone, Apexit^®^, AH Plus^TM^, Ketac^TM^ Endo, Untreated (control)	Indirect (extract)	1 d	Undiluted	1 d	Apexit^®^ < AH Plus^TM^ < Ketac^TM^ Endo < Endométhasone < N2^®^
2000	Azar et al. [55]	AH 26^®^, AH Plus^TM^, ZOE, Distilled water (positive control)	Indirect (extract)	1 h, 4 h, 8 h, 1 d, 2 d, 5 d, 1–5 w	Undiluted	22 h	AH Plus^TM^ only toxic in early phase (4 h). AH 26^®^ toxic for 1 w and ZOE for 5 w.
	Huang et al. [17]	AH 26^®^, AH Plus^TM^, Medium (control)	Direct	-	-	(a) 1 d (b) 4 h, 10 h, 1 d	AH Plus^TM^ < AH 26^®^
	Schweikl and Schmalz [69]	AH Plus^TM^, Control (n/s)	Indirect (extract)	1 d	Diluted	1 d	Sealer eluted in DMSO was toxic. Sealer eluted in sodium chloride was nontoxic.

Extraction time and cell exposure time were defined as hours (h), days (d), or weeks (w). ^1^ Material setting condition defined as fresh (F) or set (S). Abbreviations: BC, BioCeramic; BP, Bioceramic Putty; DF-MSCs, dental follicle-derived adult mesenchymal stem cells; ES, Endodontic Sealer; EWT, Extended Working Time; FS, Fast Setting; GP, gutta-percha; MTA, Mineral Trioxide Aggregate; hOCs, human osteoblastic cells; n/s, non-specified; NAC, N-acetyl-L-cysteine; NCS, Nano-Ceramic Sealer; PCS, Kerr’s Pulp Canal Sealer^TM^; O.S., osteogenic supplementation (with ascorbic acid, β-glycerophosphate, and dexamethasone); RCS, Root Canal Sealer; SE, Self-Etch; SP, Sealing Paste; ZOE, Zinc Oxide-Eugenol.

#### 3.1.2. Influence of Condition and Time of Material Setting on Cytotoxicity

To understand how the set condition of the material influences cytotoxicity, we focused on studies that used both set conditions, i.e., freshly mixed and set. Comparing AH Plus^TM^ and MTA Fillapex^®^, Zhou et al. [51] showed that AH Plus^TM^ was more toxic in freshly mixed conditions but less toxic after setting. This decrease in cytotoxicity with setting has also been confirmed by other authors [61,66,75,83]. Similarly to AH Plus^TM^, Donadio et al. [87] showed that AH 26^®^ was considerably more cytotoxic in freshly mixed conditions compared to set conditions (72 h after preparation).

#### 3.1.3. Influence of Sealer Concentration on Cytotoxicity

In order to evaluate the influence exerted by the amount of sealer on cytotoxicity, we focused on studies that used an indirect contact testing methodology with several concentrations of sealer extract. In fact, the concentration dependency of the cytotoxic effect was demonstrated for Activ GP^TM^ (Brasseler, Savannah, USA) [87], AH Plus^TM^ [51,54,71,74,78,83,91], AH 26^®^ [59,60,79,80,81,87,91], BioAggregate^®^ and iRoot^®^ SP (Innovative BioCeramix Inc., Vancouver, Canada) [85], Canals (Showa Pharmaceutical Co., Tokyo, Japan) [59,60], Endofill [78], EndoREZ^®^ [54,83], Endosequence BC^TM^ [74,76], Epiphany^®^ [28], MTA Fillapex^®^ [51,71,76], N2^®^ [59,60,80,81], ProRoot^®^ ES (Dentsply Tulsa Dental, Tulsa, USA) [74], RealSeal^TM^ (SybronEndo, Orange, CA, USA) [54,87], RoekoSeal (Roeko/Coltène/Whaledent, Langenau, Germany) [83], Roth’s Sealer (Roth International, Chicago, IL, USA) [74], Sealapex^TM^ [80,81], and Sealer 26 (Dentsply/Maillefer, Konstanz, Germany) [78]. Lee et al. [79] also showed a concentration-dependent cytotoxicity for UDMA.

#### 3.1.4. Influence of Exposure Time to Sealer on Cytotoxicity

To evaluate the influence of the time of exposure, we considered only studies that tested more than one cell incubation time point. Accordingly, 33 studies fulfilled this criterion, of which 13 used direct contact testing as a method of cell exposure to several materials. From the 33 studies, nine did not focus on comparing different incubation times [20,23,24,26,45,49,70,72,90].

A certain heterogeneity was observed in regard to this subject. Some studies showed cell viability recovery over time of exposure for BioRoot^TM^ RCS [61,63,66], GuttaFlow^®^ Bioseal and GuttaFlow^®^2 [64], MTA [21], MTA Fillapex^®^ [41], and MetaSEAL^TM^ (Parkell, Inc., Farmington, NM, USA) [29]. A recovery of cell viability was also denoted for PMMA after five weeks [29]. Other studies showed decreased cell viability over time of exposure for AH Plus^TM^ [17,25,32,38], AH 26^®^ [17], GuttaFlow^®^ (Roeko/Coltène/Whaledent, Langenau, Germany) [32], MTA Fillapex^®^ [40], Epiphany^®^ [28,32], Epiphany^TM^ SE (Pentron, Wallingford, CT, USA) [41], PCS Extended Working Time (EWT) and Apatite Root Sealers (Sankin Kogyo, Tokyo, Japan) [42], RealSeal XT (SybronEndo, Orange, CA, USA) [25], and Sicura Seal (Dentalica, Milano, Italy) and TopSeal^®^ (Dentsply DeTrey Gmbh, Konstanz, Germany) in direct contact [38].

Key et al. [31] showed recovery of cell viability at 24 h for Epiphany^®^ and ThermaSeal^®^ (Dentsply/Maillefer, Konstanz, Germany) when compared to 1 hour of exposure time, but a loss of viability for Sealapex^TM^. Jeanneau et al. [62] showed increased proliferation with increasing exposure time only for BioRoot^TM^ RCS, as the inverse relationship was observed for PCS. Bouillaguet et al. [34] also showed a higher cytotoxicity for PCS at a second 24 h and 1-week incubation periods, and also for RoekoSeal and EndoREZ^®^ at 1-week incubations, with all materials in fresh conditions. Furthermore, some studies showed a maintenance of cytotoxicity over time for PCS [27,29,43], RealSeal^TM^ and EndoREZ^®^ [27]. Mendes et al. [35] showed maintenance of cell viability for PCS and Endofill, which were classified as nontoxic.

Other studies that used “aged” sealers (i.e., sealer specimens immersed in culture media with renewal) also showed a general recovery of cell viability over time for AH Plus^TM^ and Endosequence BC^TM^ after six weeks [43], AH Plus^TM^ after five weeks [44], AH Plus Jet^®^ (Dentsply DeTrey Gmbh, Konstanz, Germany) after five weeks [29], and RealSeal SE^TM^ (SybronEndo, Orange, CA, USA) and MetaSEAL^TM^ over five weeks of observation [27]. In fact, these findings appear to be partially confirmed by studies that used different extraction time points.

Studies that performed cumulative extractions (i.e., same culture medium over the entire period of extraction) showed an increase in cytotoxicity over time of extraction for BioRoot^TM^ RCS, MTA Fillapex^®^, and SimpliSeal^®^ (Discuss Dental LLC, Calver City, KY, USA) [73]. Mandal et al. [52] showed increasing cell viability over time (72 h compared to 24 h) for AH Plus^TM^ but decreased for GuttaFlow^®^2 in set conditions.

On the other hand, studies that performed separate extractions (i.e., culture medium renewed after harvesting the extract from the previous time point)—which simulates periodontal ligament clearance [53]—showed a decrease in cytotoxicity over the time of extraction for several sealers (e.g., GuttaFlow^®^, AH Plus^TM^) [53,86]. Using similar extraction methods, Zhou et al. [51] showed a recovery of cell viability over time for AH Plus^TM^ but not for MTA Fillapex^®^, which showed increased toxicity in more concentrated extracts (1:2 and 1:8). Camps and About [47] showed a decrease in cytotoxicity for Sealapex^TM^ with no difference for AH Plus^TM^ using a root-dipping technique. However, these results were not confirmed by experiments with International Organization for Standardization (ISO) Standards 10993-5 in the same study, as only Cortisomol^TM^ (Pierre Rolland, Merignac, France) had decreasing cytotoxicity over time in this technique. Azar et al. [55] showed a decrease in cytotoxicity for both AH Plus^TM^ (only toxic in first four hours) and AH 26^®^ (toxicity decreased after one week), as no decrease was observed for ZOE cement (Produits Dentaires, Vevey Switzerland). Other studies did not compare different extraction time points or did not show significant differences [46,48,67,87].

### 3.2. In Vivo Biocompatibility

The general characteristics of the included studies are presented in Table A3 (Appendix D). As can be seen, the main reported methods were subcutaneous tissue response to sealer implants [92,93,94,95,96,97,98,99,100,101,102,103,104] and periapical tissue response to root canal filling procedure [105,106,107,108,109,110]. Specifically, in relation to root filling procedures, these were carried out primarily in premolars (both maxillary and mandibular) and also in maxillary incisors in some studies. The alveolar socket-implantation method following tooth extraction was reported by Cintra et al. [111]. Furthermore, Assmann et al. [112] studied the bone tissue response to intraosseous sealer implants in the femur of Wistar rats.

In regard to setting condition, most studies used the materials in a freshly mixed state, except for Garcia et al. [93], who only used materials in a set condition after photoactivation. Campos-Pinto et al. [96] and Derakhshan et al. [104] used materials in both freshly mixed and in set conditions. In terms of an in vivo model, rats of different species or strains were used in 12 studies and dogs were used in seven studies. In one study, New Zealand rabbits were used as animal model.

#### 3.2.1. Inflammatory Tissue Reaction to Sealers

All studies showed a generalized inflammatory response to the materials tested, as presented in Table 3. AH Plus^TM^, EndoREZ^®^, and Epiphany^®^ were the most studied sealers. Relative to the epoxy resin-based sealer AH Plus^TM^, Oliveira et al. [94] reported a nonspecific chronic inflammatory response, which can be reduced with the addition of calcium hydroxide. A slight to moderate inflammatory reaction was also reported by other authors [109]. A similar inflammatory infiltrate was shown in comparison with silicone-based sealers RoekoSeal [104,109,110] and GuttaFlow^®^2 [92], although higher comparing to GuttaFlow^®^ Bioseal within eight days of exposure [92]. Nevertheless, the same study showed that this difference had disappeared after 30 days. Assmann et al. [112] showed a lower neutrophil infiltrate in comparison to MTA Fillapex^®^, even though both sealers provided the re-establishment of original bone structure.

In regard to the methacrylate resin-based sealer Epiphany^®^, Garcia et al. [93] showed that the addition of its self-etch primer decreases the inflammatory reaction to the Epiphany/Resilon system. Similarly, Campos-Pinto et al. [96] showed that photoactivated Epiphany^®^ without primer induced a moderate to severe inflammatory reaction with extensive necrosis, whereas only slight chronic inflammatory reaction was observed in the presence of the primer. Tanomaru-Filho et al. [109] showed a slight to moderate inflammatory reaction of Epiphany^®^ comparable to AH Plus^TM^ and RoekoSeal, as Onay et al. [98] showed an inflammatory reaction that varied from none to severe at first-week observation to a none to slight reaction at the eighth-week observation. Comparing the Epiphany/Resilon with a system of PCS/Gutta-percha, Brasil et al. [107] showed similar biocompatibility, as both elicited a mild inflammatory reaction.

Concerning EndoREZ^®^, Suzuki et al. [108] showed a mild to severe inflammatory reaction. A severe tissue reaction was also shown by Zmener et al. [95] for EndoREZ^®^ combined with an accelerator (ACC, Ultradent Products Inc., South Jordan, UT, USA), by Zmener [101] in a 10-day observation and by Zafalon et al. [97], who showed high toxicity and late hypersensitive reaction to this sealer. All other materials elicited an inflammatory tissue reaction of variable degree, e.g., Sealapex^TM^ [113] and MTA Fillapex^®^ [103].

#### 3.2.2. Time of Exposure Influence on Biocompatibility

In order to understand how the time of exposure influences the biocompatibility of root canal sealers, we focused on studies that reported multiple exposure time points. Based on the included studies, time-dependency (i.e., resolution of tissue reaction over time) has been shown for the following sealers: AH Plus^TM^ [92,112], Endométhasone (Septodont, Saint-Maur-des-Fossés, France) [97], GuttaFlow^®^2 [92], GuttaFlow^®^ Bioseal [92], and RealSeal^TM^ [95]. The decrease in tissue reaction has also been shown for other materials [100].

For Epiphany^®^, contrary evidence was found, as Garcia et al. [93] and Onay et al. [98] showed a decrease in tissue reaction over time whereas Campos-Pinto et al. [96] suggested a resolution of the tissue reaction. Studies on EndoREZ^®^ also showed conflicting results, as the time-dependency was shown either as isolated sealer [101] or associated with an accelerator [95], whereas Zafalon et al. [97] showed evidence of severe inflammatory infiltrate even 90 days after implantation. Furthermore, Assmann et al. [112] showed a resolution of tissue reaction to MTA Fillapex^®^ over time, whereas Zmener et al. [103] showed maintenance of toxicity after 90 days.

#### 3.2.3. Influence of Apical Limit of Root Canal Filling on Biocompatibility

Three studies aimed to evaluate the influence of the apical limit for root canal filling on biocompatibility to root canal sealers [106,108]. Both the studies demonstrated better biocompatibility with root canal filling short of the apical foramen, in comparison with overfilling for the tested sealers, i.e., Endométhasone and EndoREZ^®^.

### 3.3. Risk of Bias

The results of the quality assessment of the studies are presented in Appendix A (in vitro) and Appendix A (in vivo) and are schematically represented in Figure 2 (in vitro) and Figure 3 (in vivo).

Regarding in vitro studies, only three studies [29,43,44] reported calculation of the sample size. Relative to the randomization process, only one study [46] reported these items. No studies reported researcher blinding to the procedures. Only a few studies reported the estimated size of effect and its precision. All studies reported information relative to the background and aims, except for two [24,82].

Concerning in vivo studies, the allocation sequence generation was unclear in several studies. No study reported allocation concealment, random animal housing, and caregiver and/or researcher blinding. Only one study [112] reported random outcome assessment. Other sources of risk of bias were found in most of the studies, mainly due to unit of analysis errors (e.g., multiple interventions per animal) and due to the addition of animals to replace drop-outs from the original sample.

## 4. Discussion

In the context of root canal therapy, materials used for root canal filling may come into contact with the periapical tissue [2]. Ideally, these materials should allow or promote the resolution of periapical inflammatory and/or infectious processes, also preventing further contamination with microorganisms [2]. Among the biological properties desirably shown by sealers (e.g., antimicrobial effect, osteogenic potential), biocompatibility is considered a key property of root canal sealers [2,3,76], thus demonstrating the importance of the study of the biocompatibility of different endodontic materials [52].

For root canal filling, the combination of a sealer with a central core material, such as gutta-percha, has been a standard [2,5]. Several reasons support the widespread use of gutta-percha, namely its plasticity, low toxic potential, ease of manipulation, radiopacity, and ease of removal, even though the lack of adhesion to dentin and shrinkage after cooling are known disadvantages of this material [2]. Other core materials and/or obturation systems have also been developed, such as resin-based obturation systems with the high-performance synthetic polyester-based Resilon (e.g., in association with RealSeal^TM^ or Epiphany^®^) and Activ GP^TM^, which consists of glass ionomer-impregnated gutta-percha cones [2,5].

Here, we aimed to perform a systematic review of the literature on the in vitro cytotoxicity and in vivo biocompatibility of root canal sealers in order to understand how these materials (individually or by type) perform in experimental cell and animal models. The inclusion of both types of study allowed a more complete perspective on the biocompatibility of these materials to be presented, as it includes information on both direct cellular toxicity (in vitro) and inflammatory tissue reaction (in vivo). Furthermore, we also aimed to understand how the material setting condition, concentration, time, and type of exposure influence the cytotoxicity and biocompatibility of these materials. As a multiplicity of methods and conditions has been reported in previous studies in this area, an overview on this subject could become difficult as well as the interpretation of the results. Therefore, a systematic review of the literature may be a useful tool to integrate such concepts and data.

Previous systematic reviews have focused primarily on the superior properties of calcium silicate-based sealers in comparison with conventional materials [12,13,14]. Here, we included all types of endodontic sealer and aimed at comparing in vitro and in vivo evidence.

Over the years, several materials have been developed for root canal filling. According to chemical composition and structure, sealers may be classified into the following types: zinc oxide-eugenol-based, resin-based, glass ionomer-based, silicone-based, calcium hydroxide-based, and bioactive endodontic sealers. AH Plus^TM^ has been the most studied sealer over the last two decades, either as a test sealer or as reference material, and thus we applied a date limit to our search in order to retrieve articles since its introduction as a new substitute to AH 26^®^ [17].

In this systematic review, the set of included studies assessed the cytotoxicity and biocompatibility of multiple sealers of the different types. Among in vitro studies, the most studied sealers were the zinc oxide-eugenol-based PCS, the epoxy resin-based AH 26^®^ and AH Plus^TM^, the methacrylate resin-based EndoREZ^®^ and Epiphany^®^, the calcium hydroxide-based Sealapex^TM^ and the bioactive sealers Endosequence BC^TM^ and MTA Fillapex^®^. AH Plus^TM^, EndoREZ^®^ and Epiphany^®^ were also the most studied in vivo.

Concerning in vitro cytotoxicity, the results suggested lower cytotoxic potential from bioactive sealers, even though some conflicting evidence was found, particularly in regard to MTA Fillapex^®^, which may be due to the release of lead in set conditions [51]. This lower in vitro cytotoxicity of bioactive sealers is in accordance with previous systematic reviews on the biological, physiochemical, and clinical properties of calcium silicate-based sealers in comparison with conventional materials [12,13,14].

Considerable methodological heterogeneity was observed in relation to several parameters, for example material setting condition, setting time, and sealer extract concentration. As for setting condition, several studies performed experiments with freshly mixed sealers; others used set materials, while some others used both freshly mixed and set conditions. Moreover, multiple setting times were reported from 1 hour to 1 month. In general, studies that assessed both conditions reported a differential in cytotoxic potential, with freshly mixed materials exhibiting higher cytotoxic potential.

The important role of setting conditions on the biological properties of sealers has been recognized, as differences have been reported between fresh and set sealers [20,21], which may account for some of the heterogeneity in the literature. However, such differences seem to decrease with setting [2,10]. The release of unconverted monomers may play a role in the cytotoxicity of sealers in freshly mixed conditions, whereas in set conditions, a residual toxic effect that is amount- and elution kinetics-dependent may be expected for these compounds [25]. However, the leaching of unconverted or partially converted constituents with potential toxicity may also remain after the setting of the material [22]. In fact, the role of setting time has been studied by Camargo et al. [83], who suggested that further research should be carried out aiming at evaluating this setting time-dependency over longer experimental periods. Arun et al. [22] also suggested that long-term clinical studies are important to understand if these materials maintain as cytotoxic over time or lose their initial toxic potential.

From a clinical perspective, the use of freshly mixed sealers is relevant because these materials are applied in an unset condition when introduced in root canals, coming into contact with the periapical tissues [25,45].

In studies that tested sealer extracts in multiple concentrations, the results suggested a concentration-dependency of the cytotoxicity, i.e., increasing cytotoxic potential with increasing extract concentration, for several sealers.

Furthermore, different contact methods were used, specifically direct contact testing, indirect contact testing with sealer extracts, and indirect contact testing with the incubation of cells with sealer specimens (without direct contact, using inserts, for example). Previous studies have suggested that direct contact exposure may lead to increased toxicity, in comparison with other methods and in spite of the acceptable clinical performance [27,47]. However, as direct contact between the sealer and the periapical tissue is possible during and after root filling procedures [2,47], such a contact method may provide important information on the cytotoxicity of these materials, as it simulates the possible extrusion of unset sealer in the periapical tissues [25,27,29,47]. Furthermore, some studies used root models [45,46,47,48], which may represent a useful model as this attempts to simulate the reality of endodontic treatments [45].

Regarding the influence of exposure time on the cytotoxic effects of the materials, studies showed a certain heterogeneity. Interestingly, studies that used washed-out or “aged” sealers reported a general recovery of cell viability over five to six weeks of observation [27,29,43,44].

As mentioned, these findings seem to be supported by studies that tested sealers by extraction methods, with different extraction time points. Methodologically, a difference in studies was observed in this regard, as some studies performed cumulative extractions, i.e., no medium renewal, and others carried out separate extractions—that is, with medium renewal. In general, the first method appears to be related to higher cytotoxic effects. In a way, such findings may be related to the time-dependent release of compounds with setting, as previously discussed.

The in vitro studies included in this systematic review are indicative of differences between the various root canal sealers. Furthermore, most studies followed the ISO Standards 10993-5:2009, which encompass direct and indirect contact methods, fresh and set materials, and several extract concentrations. However, the concentrations tend to be higher compared to the clinical context. Therefore, care should be taken when extrapolating these results for clinical practice.

In addition, high heterogeneity was observed regarding the cell model used for cytotoxicity assessment, from stem cells of different origins to osteoblasts or fibroblasts, as previously acknowledged for MTA [114].

Relative to the in vivo evidence, all studies showed an inflammatory reaction in response to the various sealers, independently of type, ranging from slight to severe inflammatory reactions. Nevertheless, studies also generally suggested that the tested sealers presented acceptable biocompatibility. The ability to provide the re-establishment of original bone structure was also shown for some sealers, such as AH Plus^TM^ and MTA Fillapex^®^ [112].

Moreover, different methods were used for the assessment of tissue response to sealers. The ISO Standard 7405 on the biological evaluation of dental materials was followed in most studies. In this context, several studies assessed the tissue response to subcutaneous or intraosseous sealer implantation, and others assessed the periapical tissue response to root filling procedure.

In one study [111], a method of implantation in the alveolar socket post-extraction was reported. Of the studies that evaluated periapical tissue response after root filling procedures, Tanomaru-Filho et al. [113] carried out root-end filling procedures after periapical lesion induction in order to simulate the clinical conditions of endodontic surgery. As both these methods may provide a more accurate representation of the clinical environment, they represent interesting approaches that could be relevant to the study of the biocompatibility of dental materials, especially endodontic materials.

The influence of exposure time on biocompatibility was shown by several studies, which showed that the initial inflammatory reaction tends to subside over time [92,95,97,100,112]. However, conflicting results were found for some sealers, specifically Epiphany^®^ and EndoREZ^®^.

In addition, two in vivo studies tested the biocompatibility of root canal fillings by the comparison of two apical limits, short of the apical foramen and overfilling [106,108]. As expected, better biocompatibility was shown in fillings short of the apical foramen.

The high risk of bias of the studies included in this systematic review represents a key limitation as well as the methodological heterogeneity, which has also been acknowledged in previous systematic reviews [13,14]. In fact, eligible studies exhibited a considerable risk of bias, with several studies lacking information on randomization processes, blinding, and outcome measures, thus highlighting the need for well-designed and well-reported preclinical and clinical studies.

## 5. Conclusions

In this study, we carried out a systematic review of the literature on the direct cellular toxicity (in vitro) and inflammatory tissue reaction (in vivo) biocompatibility of root canal sealers. The main inclusion criteria were as follows: (a) in vitro cellular studies that tested direct cellular toxicity as cell viability/proliferation and (b) in vivo animal studies that evaluated biocompatibility as inflammatory tissue reaction after subcutaneous, intraosseous, alveolar socket, or root canal implantation.

A joint analysis of the included studies reveals that endodontic sealers elicit variable effects in terms of direct cellular toxicity and inflammatory tissue reaction. In terms of sealer type, bioactive sealers showed a tendency for lower in vitro direct cytotoxicity. However, this finding was not definitively confirmed by in vivo studies, as very few studies are available with these sealers. Moreover, several factors may influence the biocompatibility of these materials, particularly setting condition and time, material concentration and type of exposure, among others.

Considerable heterogeneity was observed in the evidence between in vitro and in vivo studies as well as a considerable risk of bias. Therefore, no definitive conclusion was achievable regarding which sealer or type of sealer presents the best biocompatibility.

The direct extrapolation of these results must be treated with caution due to several aspects, namely: (a) the assessment of biocompatibility was carried out in experimental models; (b) some methods do not correlate directly to the clinical reality of endodontic treatments, e.g., testing only set materials; and (c) other material properties should be taken into account, e.g., antimicrobial and physicochemical properties.

Therefore, a better understanding of the biocompatibility of endodontic sealers requires further research with precisely designed studies and accurate and complete reporting. In this context, the following methodologic considerations should be taken into account in the design phase of biocompatibility studies in order to improve the clinical applicability of results: (a) endodontic sealers are clinically used in a freshly mixed state according to the manufacturer’s instructions and thus fresh conditions are a more accurate representation of the clinical environment in an early phase of the treatment; (b) even though testing diluted sealers provides complete information in regard to the influence of sealer concentration, these materials are used undiluted in the clinic, and thus studies should focus more on testing undiluted materials; (c) direct contact methods are a more accurate representation of the clinical use of endodontic sealers, compared to indirect contact methods; (d) the use of human-derived cell lines (namely fibroblasts and osteoblasts) should be preferred for in vitro testing over animal-derived cells or others.

From a clinical perspective, our systematic review provides an overview on the biocompatibility of endodontic sealers and the main factors that may influence endodontic treatment success, from a biocompatibility standpoint.

## Figures and Tables

**Figure 1 materials-12-04113-f001:**
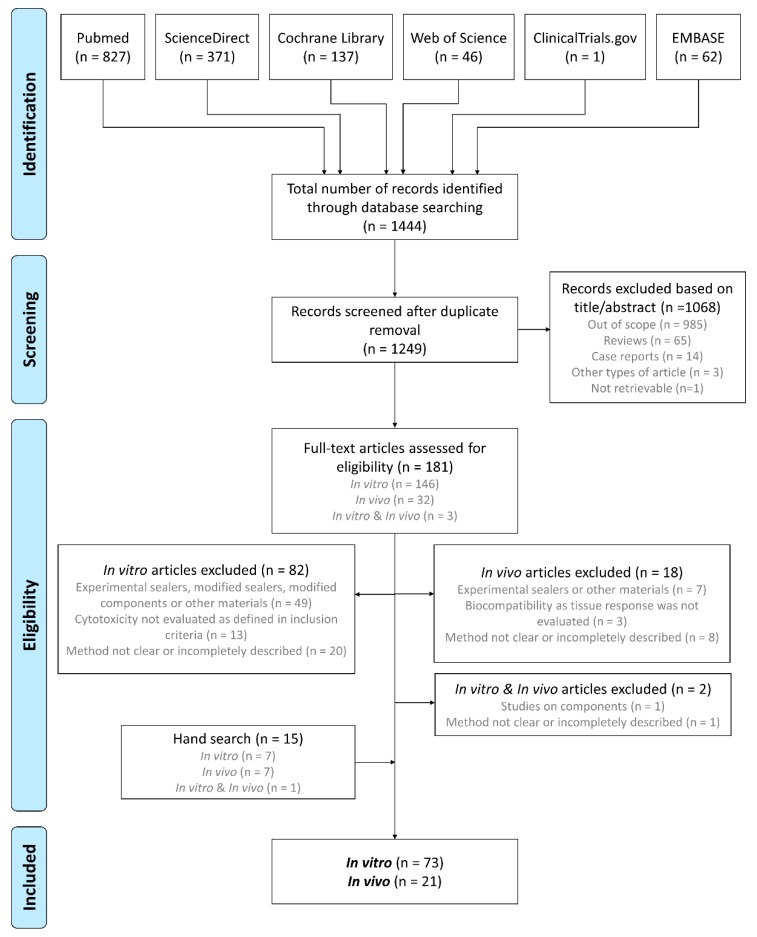
Flow diagram of identification of studies for inclusion in this systematic review according to Preferred Reporting Items for Systematic Reviews and Meta-Analyses (PRISMA) guidelines.

**Figure 2 materials-12-04113-f002:**
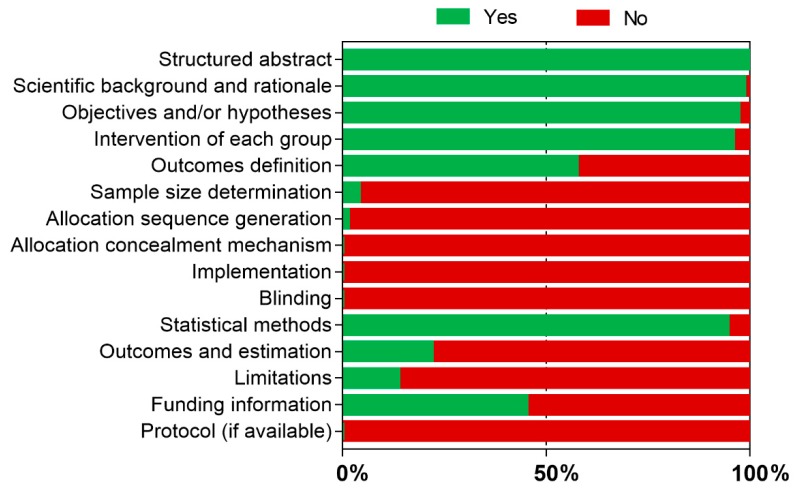
Methodological quality assessment of in vitro studies.

**Figure 3 materials-12-04113-f003:**
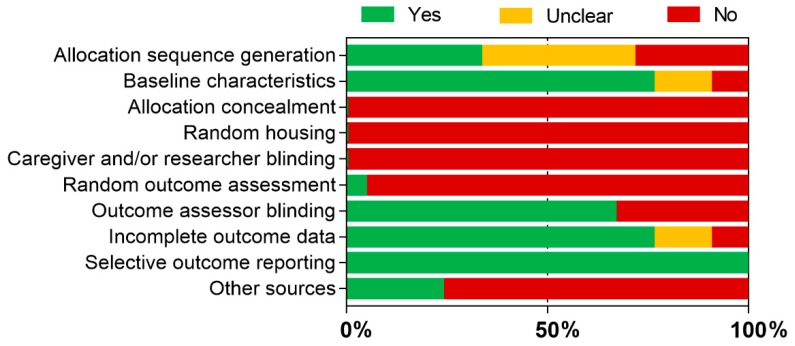
Methodological quality assessment of in vivo studies.

**Table 1 materials-12-04113-t001:** Population, Intervention, Comparison and Outcome (PICO) strategy used for assessment of scientific literature.

Parameter	Assessment
Population (P)	In vitro: cell models In vivo: animal models of tissue inflammatory reaction
Intervention (I)	In vitro: sealer specimens or sealer extracts In vivo: sealer implants (subcutaneous, alveolar socket, or intraosseous) or root filling procedures
Comparison (C)	Other root canal sealers or non-exposed control groups
Outcome (O)	In vitro: cytotoxicity (measured as cell viability or proliferation) In vivo: biocompatibility (measured as tissue response to the material)

**Table 3 materials-12-04113-t003:** Summary of parameters and results collected from included in vivo studies, ordered by publication date (from most recent).

Year	Study	Groups (G)	Tissue Response	Exposure Time	Type of Analysis	Outcomes Assessed	Biocompatibility
2019	Santos et al. [92]	G1: Empty polyethylene (PE) tube (control); G2: GuttaFlow^®^ Bioseal; G3: GuttaFlow^®^2; G4: AH Plus^TM^	Subcutaneous	8 d, 30 d	Histology (Hematoxylin-eosin, H&E)	Macrophage infiltrate, thickness of fibrous capsule, vascular changes	At 8 d, GuttaFlow^®^ Bioseal had lower inflammatory reaction than GuttaFlow^®^2, AH Plus^TM^. All biocompatible at 30 d.
2015	Assmann et al. [112]	G1: Mineral Trioxide Aggregated (MTA) Fillapex^®^; G2: AH Plus^TM^; G3: Empty cavity (control)	Bone	7 d, 30 d, 90 d	Histology (H&E)	Inflammatory infiltrate, fibers and hard tissue barrier formation	Both sealers provided re-establishment of original bone tissue structure. Inflammatory reaction decreased over time.
2014	Silva et al. [105]	G1: Sealapex Xpress^TM^/GP (Gutta-Percha); G2: RealSeal XT/Resilon	Periapical	90 d	Histology (H&E and immunohistochemistry or IHC for mineralization markers)	Biological apical sealing, inflammatory infiltrate, root and bone resorption	Both sealers allowed biological apical sealing with deposition of mineralized tissue.
2012	Zmener et al. [103]	MTA Fillapex^®^; Grossman’s sealer (positive control)	Subcutaneous	10 d, 30 d, 90 d	Histology (H&E)	Thickness of a fibrous capsule, vascular changes, and various types of inflammatory cells	MTA Fillapex^®^ toxic for 90 d, Grossman’s sealer toxic only at 10 d and 30 d
2011	Suzuki et al. [106]	G1: Endométhasone/GP (short of apical foramen); G2: Endométhasone/GP (overfilling)	Periapical	90 d	Histology (H&E)	Biological apical sealing, root resorption, inflammatory infiltrate, presence of giant foreign-body cells and thickness and organization of periodontal ligament (PDL)	Chronic inflammatory infiltrate in all specimens. Best result obtained with filling short of the apical foramen (vs. overfilling).
2010	Garcia et al. [93]	Epiphany/Resilon (G1: with self-etch primer, G2: without primer); G3: Endofill/GP; G4: Empty dentin tube	Subcutaneous	7 d, 21 d, 42 d	Histology (H&E)	Inflammatory infiltrate, capacity of cellularity and vascularization, macrophagic activity	Epiphany/Resilon system with primer had lower inflammation, compared to system without primer, but higher compared to Endofill + GP.
	Oliveira et al. [94]	G1: AH Plus^TM^; G2: AH Plus^TM^ with calcium hydroxide 5% (w/w); G3: Control (n/s)	Subcutaneous	14 d	Histology (H&E, Masson´s Trichrome)	Inflammatory response (lymphocytes, plasmocytes, neutrophils, eosinophils, macrophages, giant foreign-body cells, blood vessels)	All showed nonspecific chronic inflammation. Calcium hydroxide improved biocompatibility of AH Plus^TM^.
	Brasil et al. [107]	G1: Epiphany^®^/Resilon system; G2: Kerr’s Pulp Canal Sealer^TM^ (PCS)/GP	Periapical	60 d	Radiographic evaluation and histology (H&E)	Radiographic evaluation (quality of filling, apical limit and extruded material) and histology (biological apical sealing, PDL thickness, inflammatory reaction, resorption)	Similar biocompatibility between systems: mild inflammatory reaction (macrophages and lymphocytes).
	Zmener et al. [95]	G1: EndoREZ^®^ + polymerization accelerator; G2: RealSeal^TM^; G3: PCS (positive control); G4: Solid silicone rods (control)	Subcutaneous	10 d, 30 d, 90 d	Histology (H&E)	Fibrous capsule formation, inflammatory infiltrate (polymorphonuclear or PMN leukocytes, lymphocytes, plasmocytes, macrophages, giant foreign-body cells), capillaries	EndoREZ^®^ and RealSeal^TM^ had severe inflammation reaction (resolved over time). PCS had severe reaction (over time).
	Suzuki et al. [108]	G1: EndoREZ^®^/GP (short of the apical foramen); G2: EndoREZ^®^/GP (overfilling)	Periapical	90 d	Histology (H&E, Brown and Brenn staining)	Biological apical sealing, apical cementum resorption, intensity of inflammatory infiltrate, organization and thickness of PDL	Both groups showed inflammation. Best result obtained with filling short of the apical foramen (vs. overfilling).
2009	Tanomaru-Filho et al. [109]	G1: Intrafill; G2: AH Plus^TM^; G3: RoekoSeal; G4: Epiphany^®^/Resilon system	Periapical	90 d	Histology (H&E, Mallory Trichrome)	Intensity of inflammatory infiltrate, PDL thickness, bone and apical cementum resorption, biological apical sealing	AH Plus^TM^, RoekoSeal, Epiphany^®^ (slight to moderate) > Intrafill (severe inflammation and PDL thickening)
	Derakhshan et al. [104]	RoekoSeal Automix, AH 26^®^, AH Plus^TM^, Empty PE tubes (control)	Subcutaneous	7 d, 14 d, 60 d	Histology (H&E)	Thickness of connective tissue capsule, severity and extent of inflammation and necrosis	RoekoSeal and AH Plus^TM^ biocompatible; extent of inflammation was higher with AH26^®^
2008	Leonardo et al. [110]	G1: RoekoSeal Automix; G2: AH Plus^TM^	Periapical	90 d	Histology (H&E, Mallory Trichrome, Brown and Brenn staining)	Newly mineralized formed tissue, periapical inflammatory infiltrate, apical PDL thickness, cementum, dentin and bone resorption	For biological apical sealing: RoekoSeal > AH Plus^TM^. Similar infiltrate, PDL thickening and resorption.
	Campos-Pinto et al. [96]	G1: Epiphany^®^; G2: Photoactivated Epiphany^®^; G3: Epiphany^®^ with self-etch primer; G4: Photoactivated Epiphany^®^ with primer; G5: Empty PE tube	Subcutaneous	7 d, 21 d, 42 d	Histology (H&E)	Neutrophils, leukocytes, macrophages, lymphocytes, plasmocytes, giant foreign-body cells, dispersed material, necrotic tissue	All groups showed mild inflammation. Group 2 showed necrosis and more inflammation.
2007	Zafalon et al. [97]	G1: Endométhasone; G2: EndoREZ^®^ (lateral wall outside of Teflon tube was the negative control)	Subcutaneous	15 d, 30 d, 60 d, 90 d	Histology (H&E)	Féderation Dentaire Internationale (FDI) criteria: new bone, neutrophils, macrophages, lymphocytes, plasmocytes, giant foreign-body cells, dispersed material, capsule, necrotic tissue, resorption	Endométhasone (tissue reaction decreased over time) > EndoREZ^®^ (highly toxic and late hypersensitive reaction)
	Onay et al. [98]	G1: Teflon (negative control); G2: Epiphany^®^; G3: Gutta-percha; G4: Resilon	Subcutaneous	1 w, 4 w, 8 w	Histology (H&E, Masson´s Trichrome)	Stromal inflammatory response, infiltration of mast cells, proliferation of fibroblasts, vascular changes, granulation tissue, giant foreign-body cells	All groups induced inflammation. Tissue reaction decreased over time.
2006	Tanomaru-Filho et al. [113]	G1: Sealer 26; G2: Sealapex^TM^ + ZnO (Zinc Oxide); G3: MTA; G4: No retrofilling	Periapical (after lesion)	180 d	Histology (H&E, Mallory Trichrome)	Periapical inflammatory infiltrate, apical PDL thickness, deposition of cementum on the sectioned apical surface, cementum and bone resorption, apical dentin resorption	Sealer 26, Sealapex^TM^ with ZnO and MTA provided periapical repair. Control showed unsatisfactory periapical repair.
	Cintra et al. [111]	G1: Empty PE tubes (control); G2: ProRoot^®^ MTA; G3: MBPc (new calcium hydroxide-based sealer)	Alveolar	7 d, 15 d, 30 d	Histology (H&E, Brown and Brenn staining)	Extent and intensity inflammatory infiltrate based on cell count and extension beyond implants	All groups showed similar biological response (mild to moderate inflammatory response).
2004	Kim et al. [100]	G1: PCS EWT; G2: Apatite Root Sealer (ARS) type I; G3: ARS type II; G4: CAPSEAL I; G5: CAPSEAL II; G6: Empty polytetrafluoroethylene (PTFE) tube (control)	Subcutaneous	1 w, 2 w, 4 w, 12 w	Histology (H&E)	Thickness of reaction zone, inflammatory infiltrate (macrophages, plasmocytes, lymphocytes, neutrophils	Capseal groups showed lower tissue response than others. In all groups, inflammatory reaction decreased over time.
	Zmener [101]	G1: EndoREZ^®^; G2: Solid silicone rods	Subcutaneous	10 d, 30 d, 90 d, 120 d	Histology (H&E)	Fibrous capsule formation, inflammatory infiltrate (PMN leukocytes, lymphocytes, plasmocytes, macrophages, giant foreign-body cells), capillaries	Inflammation was observed with EndoREZ^®^ (decreased with time). Control showed mild inflammation only at 10 d.
2001	Figueiredo et al. [102]	G1: *N*-Rickert; G2: AH 26^®^; G3: Fillcanal; G4: Sealer 26	Subcutaneous	90 d	Histology (H&E)	Histopathologic evaluation (granulation tissue, lymphocytes, PMN neutrophils and eosinophils, plasmocytes, macrophages, giant foreign-body cells)	Sealer 26 (mild irritation) > *N*-Rickert and AH 26^®^ (moderate) > Fillcanal (severe irritation).

N represents the number of animals in studies with implantation methods or the number of root canals in studies with root canal filling procedures. Exposure time was defined in days (d) or weeks (w). Abbreviations: ARS, Apatite Root Sealer; FDI, Féderation Dentaire Internationale; GP, Gutta-percha; H&E, Hematoxylin-eosin; IHC, immunohistochemistry; MTA, Mineral Trioxide Aggregate; n/s, non-specified; PA, periapical; PCS, Kerr’s Pulp Canal Sealer^TM^; PE, polyethylene; PDL, periodontal ligament; PMN, polymorphonuclear; PTFE, polytetrafluoroethylene; ZnO, zinc oxide.

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
