# Peer review of "Biocompatibility of Root Canal Sealers: A Systematic Review of In Vitro and In Vivo Studies"

_materials, 2019, doi:10.3390/ma12244113_

Round 1

Reviewer 1 Report

Dear authors 

Thank you for the opportunity to read and review the present manuscript. The manuscript is very interesting and presents sound scientific data about biocompatibility of root canal sealers. I am very impressed with the work you have done and I believe that it will be a significant contribution to the endodontic literature

Author Response

Reviewer: Dear authors, Thank you for the opportunity to read and review the present manuscript. The manuscript is very interesting and presents sound scientific data about biocompatibility of root canal sealers. I am very impressed with the work you have done and I believe that it will be a significant contribution to the endodontic literature.

Our reply: We appreciate the interest and thank the kind comments regarding our manuscript.

Reviewer 2 Report

This is a very well-written review manuscript, which is very helpful for readers to understand the biocompatibility of commercial root canal sealers.

However, the Conclusions section might be discouraging for dental clinicians, who want to obtain valuable clinical information from this article. In the Conclusions section, please add some descriptions about the selection criteria and appropriate handling guidelines of the materials.

Author Response

Reviewer: This is a very well-written review manuscript, which is very helpful for readers to understand the biocompatibility of commercial root canal sealers. However, the Conclusions section might be discouraging for dental clinicians, who want to obtain valuable clinical information from this article. In the Conclusions section, please add some descriptions about the selection criteria and appropriate handling guidelines of the materials.

Our reply: We appreciate the interest and are thankful for the kind and constructive comments. According to the referee’s recommendation, the conclusion of our manuscript was rewritten, in order to improve and clarify the clinical applicability of the concepts discussed in this manuscript.

Reviewer 3 Report

Dear authors, 

thank you very much for your paper.

In this paper the authors presented  a systematic review of the literature  (2000-2019) on the in vitro cytotoxicity and in vivo biocompatibility of root canal sealers, in order to understand how these materials  perform in terms of biocompatibility in experimental cell and animal models.

Actually, it is well organized and well written.

However, there are a lot of narrative reviews in the literature, why you did not perform a meta-analysis too? It should have improved the manuscript.

I should suggests a few changes in the manuscript:  

Please better specify the period of collection. You mentioned that the period is 2000-2019. You inserted the review in PROSPERO  in July. Please specify the months of study inclusions, as it is not correct to say 2019 and include some studies... doing so you may have lost some studies.  please correct it. 

please review the English language. there are few mistakes in the text. 

please complete the data on PROSPERO register (as actually a lot of data are missing)

line 49: you speak about Grossman criteria... why you mention also the reference n.2? Could you please eliminate ref. 2 and let only 3 in this context? otherwise you should not speak about Grossman's criteria.

line 57-59: please control the English language. 

please review all the tex for typos...e.g (line 346)

Author Response

Reviewer: Dear authors, thank you very much for your paper. In this paper the authors presented a systematic review of the literature (2000-2019) on the in vitro cytotoxicity and in vivo biocompatibility of root canal sealers, in order to understand how these materials perform in terms of biocompatibility in experimental cell and animal models. Actually, it is well organized and well written. However, there are a lot of narrative reviews in the literature, why you did not perform a meta-analysis too? It should have improved the manuscript.

Our reply: We appreciate the interest and thank the kind comments regarding our manuscript. Indeed, there are several narrative reviews and also systematic reviews on this subject, as detailed in our manuscript. In general, such systematic reviews have had a more focused nature, as they have explored the potential superiority in the in vitro biocompatibility of calcium silicate-based sealers. However, we intended in our manuscript to present a more global, general and inclusive perspective on the biocompatibility of the several types of endodontic sealers that have been commercially available as well as both in vitro and in vivo evidence. As highlighted in our manuscript, the existing literature on this subject exhibits an inherent heterogeneity of methodology and reporting, thus we aimed to perform a systematic review without meta-analysis.

Reviewer: I should suggests a few changes in the manuscript: Please better specify the period of collection. You mentioned that the period is 2000-2019. You inserted the review in PROSPERO  in July. Please specify the months of study inclusions, as it is not correct to say 2019 and include some studies... doing so you may have lost some studies.  please correct it.

Our reply: According to the referee’s recommendation, we revised the manuscript and specified the period of collection, between 2000 and 2019 (June 11), both in the abstract and throughout the text.

Reviewer: Please review the English language. there are few mistakes in the text.

Our reply: We revised the English language in our manuscript. Furthermore, the manuscript was also revised by a professional proofreader and native speaker of English, who has attested that our manuscript is now written in correct and clear English language, as evidenced by the documentation attached to this revision.

Reviewer: Please complete the data on PROSPERO register (as actually a lot of data are missing)

Our reply: Accordingly, we have completed the data on the PROSPERO record. The updated version will be published once the updated information has been checked by PROSPERO.

Reviewer: Line 49: you speak about Grossman criteria... why you mention also the reference n.2? Could you please eliminate ref. 2 and let only 3 in this context? otherwise you should not speak about Grossman's criteria.

Our reply: We revised the text in order to include only the original reference for the Grossman’s criteria.

Reviewer: Line 57-59: please control the English language.

Our reply: The referred text has been rewritten and corrected.

Reviewer: Please review all the tex for typos...e.g (line 346)

Our reply: According to the referee’s recommendation, we checked and corrected the entire manuscript for typos.

Reviewer 4 Report

This is an extensive and thorough work. Unfortunately no solid conclusion could be drawn.

Would you please explain (in the introduction part) which was the reason for choosing to assess these particular criteria: in vitro cytotoxicity and in vivo biocompatibility? An attempt is made in the discussion part, but it does not clarify the matter.

Line 174-175, 268-273 etc. Please give the manufacturer and location for the products each time you mention.

Author Response

Reviewer: This is an extensive and thorough work. Unfortunately no solid conclusion could be drawn.

Our reply: We appreciate the constructive comments. We revised the Conclusions section and have tried to improve and clarify the clinical applicability of the concepts that were discussed in our manuscript.

Reviewer: Would you please explain (in the introduction part) which was the reason for choosing to assess these particular criteria: in vitro cytotoxicity and in vivo biocompatibility? An attempt is made in the discussion part, but it does not clarify the matter.

Our reply: As detailed in our manuscript, we aimed to perform a systematic review of the literature on the in vitro cytotoxicity and in vivo biocompatibility of root canal sealers. First, the assessment of the in vitro cytotoxicity represents a measure of direct cellular toxicity which is important to consider when studying the biocompatibility of clinically used materials. However, evidence obtained from in vitro studies presents several limitations, particularly in regard to the clinical translation of the results. Therefore, we also included in vivo studies which tested the biocompatibility as inflammatory tissue reaction to these materials. The inclusion of both types of study allowed to present a more complete perspective on the biocompatibility of endodontic sealers as well as to compare the results and understand how the evidence correlates between both types of study. Considering the question raised by the referee, we revised the introduction and discussion in order to clarify this subject.

Reviewer: Line 174-175, 268-273 etc. Please give the manufacturer and location for the products each time you mention.

Our reply: According to the referee’s recommendation, we included information on the manufacturer in the manuscript. As the various sealers are mentioned several times throughout the manuscript, we propose to include the manufacturer’s information on the first mention of the commercial brand, so as not to overload the text with manufacturer information and to make it easier to read. Nevertheless, if it is absolutely necessary to mention the manufacturer every time the brand is mentioned in the text, we will make the necessary corrections.